# Comparative Analysis of the Genetic Diversity of Chilean Cultivated Potato Based on a Molecular Study of Authentic Herbarium Specimens and Present-Day Gene Bank Accessions

**DOI:** 10.3390/plants12010174

**Published:** 2022-12-31

**Authors:** Tatjana Gavrilenko, Irena Chukhina, Olga Antonova, Ekaterina Krylova, Liliya Shipilina, Natalia Oskina, Ludmila Kostina

**Affiliations:** N.I. Vavilov All-Russian Institute of Plant Genetic Resources, Bolshaya Morskaya 42-44, 190000 Saint-Petersburg, Russia

**Keywords:** *Solanum tuberosum*, cytoplasmic genetic diversity, herbarium specimens, potato germplasm collections, plastid DNA haplotypes, microsatellites

## Abstract

At the end of the 1920s, Vavilov organized several potato-collecting missions in South and Central America. Vavilov and his colleagues, Juzepczuk and Bukasov, participated in these expeditions and worked on gathered material, designated two centers of potato varietal riches and diversity—the Peru–Bolivia high-mountain center and the southern coast of Chile. The WIR Herbarium holds authentic specimens of many taxa described by Russian taxonomists. Here, a set of 20 plastid DNA-specific markers was applied for 49 authentic herbarium specimens of *Solanum tuberosum* L. from the WIR Herbarium to analyze the genetic diversity of the landrace population collected by Juzepczuk in 1928 in southern–central Chile. Two plastid DNA types, T and A, and two chlorotypes were identified in herbarium specimens, with a clear predominance (96%) of chlorotype cpT_III. In addition, we analyzed 46 living Chilean accessions from the VIR field potato gene bank that were collected after the appearance of *Phytophthora infestans* in Chile. These living accessions were differentiated into four chlorotypes. Finding a D-type cytoplasm in living Chilean accessions that possess two new chlorotypes indicates a replacement of native cultivars and introgression from the wild Mexican species *S. demissum* that was actively used in breeding as a source of race-specific resistance to late blight.

## 1. Introduction

Section *Petota* Dumort. comprises numerous tuber-bearing species of the genus *Solanum* L. Their distribution area stretches along the western coast of two American continents, reaching the southern states of the USA in the north and southern regions of Chile and Argentina in the south [1]. The latest revision [2] reduced the number of wild potato species to 107. Section *Petota* contains several domesticated species, including *Solanum tuberosum* L. The number of cultivated potato species is disputed. Seven cultivated species were recognized in the most commonly accepted treatment of Hawkes [3], with only four cultivated species in the latest taxonomic estimate [4,5]. A tremendous diversity of upland Andean landrace populations is widely distributed from Venezuela to northern Argentina [6]. They are separated geographically by the Atacama Desert and the Andean Cordillera from the lowland Chilean landrace populations, which are concentrated in the Chiloé and Chonos Archipelagos, as well as in the adjacent mainland of Chile.

### 1.1. Historical Concepts of the Origin of Chilean Cultivated Potato

In 1929, Juzepczuk and Bukasov provided the first taxonomic treatment of cultivated potato species and the first hypothesis of their origin [7] based on the results of their expeditions between 1925 and 1933 in South and Central America that were organized by Vavilov, as well as based on systematic studies of collected potato germplasms maintained in the Vavilov’ Institute in Leningrad (now Saint Petersburg). Russian botanists designated two centers of potato varietal riches and diversity in the Peru–Bolivia plateau and in Chiloé, Southern Chile [7,8,9,10]. They proposed the ‘multiple origin hypothesis’ and assumed the polyphyletic origin of 12 Andean cultivated species in the territory of the present-day Peruvian–Bolivian–Colombian Andes and the independent domestication of Chilean potato—*Solanum tuberosum* (*sensu stricto*)— on Chiloé Island [7,8]. Juzepczuk formally described two Chiloan species, *Solanum leptostigma* Juz. and *Solanum molinae* Juz. and declared them as wild progenitors of Chilean cultivated potato [11], whereas Hawkes [3,12] considered these two species as naturalized escapes from cultivation. In 1933, Bukasov [8] proposed an intraspecific classification for Chilean and Andean cultivated potato species. Specimens of taxa named by Russian botanists are held in two herbaria, WIR and LE, both located in Saint Petersburg [5,13].

Furthermore, the question of the origin of the Chilean cultivated potato has been debated in terms of the geographic location of its domestication and of its putative progenitors. Ugent et al. [14] supported the hypothesis of the independent origin of Chilean cultivated potato and suggested that it originated through the spontaneous chromosome doubling of diploid populations of wild species *Solanum maglia* Schltdl., which ranges from the Argentinian province of Mendoza to central Chile.

According to a ‘restricted origin’ hypothesis, potatoes were first domesticated in the Central Andes, in the Lake Titicaca basin. The Andean domesticated populations were spread in prehistoric times from this primary center into different regions of South America, including Chile [12,15,16,17,18]. According to this hypothesis, the Chilean cultivated potato originated from the introduced Andean tetraploid landrace populations through selection towards adaptation to the long-day photoperiod of the coastal regions of southern Chile. Hawkes [3] classified the Chilean cultivated potato under *S. tuberosum* subsp. *tuberosum*, and Andean tetraploid landraces as *S. tuberosum* subsp. *andigena*. In the latest taxonomic treatment, the Chilean cultivated potato is recognized as the *S. tuberosum Chilotanum* group, and Andean landraces are recognized as the *S. tuberosum Andigenum* group [4,5].

Grun [19] suggested the ‘hybrid origin’ of the Chilean cultivated potato from spontaneous interspecific crosses between Andean landraces and an unidentified wild species as a female parent. This unknown species contributed cytoplasmic sterility factors to the Chilotanum germplasm [19,20,21].

### 1.2. Molecular Studies of the Origin and Genetic Diversity of Chilean Cultivated Potato

New insight into the origin of cultivated potatoes has come from molecular research; nuclear DNA (nDNA) markers and genomics provided evidence for single potato domestication in southern Peru and in adjacent northern Bolivia from a wild progenitor in the *Solanum brevicaule* Bitter complex [2,22,23,24].

Cytoplasmic genetic diversity and the maternal lineage of cultivated potato species have been investigated using DNA markers specific to different loci of plastid DNA (cpDNA) and mitochondrial DNA (mtDNA) [25,26,27,28,29,30]. Based on these data, Hosaka and Sanetomo [29] selected a set of PCR-based cytoplasmic markers for the detection of five main cpDNA types (A, C, S, T, and W), three main mtDNA types (α, β, and γ), and six basic cytoplasm types (A, M, P, T, D, and W) in cultivated potato and its wild relatives.

Plastid DNA data did not confirm a theory of the direct selection of Chilean *S. tuberosum* from tetraploid Andean landraces. More than 80% of contemporary Chilean landraces have T-type cpDNA, which is characterized by a specific 241 bp deletion in the *ndhC/trnV* intergenic spacer; the rest of the Chilean cultivars possess A- and W-type cpDNA (both without this deletion) [26,31,32,33,34]. In contrast, A-type cpDNA is predominant in Andean tetraploid landraces [26,34], and fewer than 5% of them have T-type cpDNA [4,26,33,34,35,36]. Plastid microsatellite (cpSSR) data also differentiated Chilean and Andean landraces [27,36].

A wild maternal ancestor of the Chilean cultivated potato, identified in a wide subset of potato species, was *Solanum tarijense* Hawkes (*Solanum berthaultii* Hawkes), native to southern Bolivia and northern Argentina. Approximately 20% of the analyzed accessions of this wild species have the 241 bp plastid deletion and hence, T-type cpDNA [32,37]. Based on these data, Hosaka [33] concluded that Chilean *S. tuberosum* was selected from natural hybrids between populations of *S. berthaultii* exhibiting the 241 bp deletion as females and Andean landraces.

Genomics approaches further advance a ‘hybrid origin’ hypothesis and provide evidence that Chilean landrace populations derived from hybridization events between short-day adapted members of the *S. tuberosum Andigenum* group and wild species such as *Solanum microdontum* Bitter native to Bolivia and northern Argentina [38]. This wild species could be a donor of the adaptive alleles allowing hybrid populations to form tubers under the long day conditions of southern Chile [38].

Further migration of these long-day-adapted hybrid populations (pre-Chilotanum) to Chile could be accompanied by new hybridization events. Thus, GBSSI sequencing data revealed that *S. maglia* and Chilean *S. tuberosum* have the same waxy gene alleles [39]; nuclear SSRs grouped all analyzed accessions of these two species together [40]. However, cpDNA data do not support the hypothesis of Ugent [14] concerning the direct origin of Chilean cultivated potato from *S*. *maglia* because T-type cpDNA was not found in the analyzed samples of this wild species. The same cpSSR haplotype #II was found in both *S. maglia* and most Andean tetraploid landraces [36]; thus, it has been suggested that Chilotanum was formed by hybridization between *S. maglia* as a possible paternal ancestor and pre-Chilotanum as a maternal ancestor [36,40].

There are still many gaps in understanding the genetic diversity of Chilean landrace populations over time. The literature documents losses in Chilean landrace diversity in the second half of the 20th century after the appearance in Chile of *Phytophthora infestans* (Mont.) De Bary, which causes late blight disease. The first confirmed record of potato blight in Chile was in October 1950; in 1951, the Chilean crops suffered the first devastating epidemic [41,42,43]. Cox and Large [41] also pointed to the earlier introduction of *P. infestans* sometime between 1937 and 1948 and considered that it probably came from Argentina. A number of well-known potato taxonomists and germplasm collectors, such as Dodds, Ochoa, Contreras, Spooner, and Zykin, who visited Chile after this late blight epidemic, indicated that most Chilean landraces which were extremely susceptible to late blight were no longer cultivated, particularly in the Chiloé region, and replaced by European and North American commercial varieties which had some field resistance to the *P. infestans* [41,44,45,46,47].

Reports of germplasm-collecting missions document a progressive increase in the number of commercial varieties introduced to southern–central Chile, especially after the 1940s. For example, two missions (both organized by the USDA) led by Husbands in 1908 and by Erlanson and MacMillan in 1932 each gathered approximately 150 indigenous cultivars. In their reports, there were no more than 3% of “probably introduced stocks” noted by them as single European varieties growing at that time in Chile [48,49]. Since 1951, there have been extensive substitutions of a mixture of indigenous cultivars for European varieties [41]. Later, in 1990, members of the international germplasm-collecting expedition noted that European introductions and new varieties predominated in Chiloé, whereas the indigenous landraces maintained in small plots on farms in most cases were virus-infected [47].

Molecular data also traced changes in the genetic diversity of Chilean landrace populations. The fact that about 20% of the contemporary Chilotanum landraces lack the 241 bp plastid deletion [4,26,35,36,50] may reflect genetic erosion caused by the human transmission of European varieties to Chile. Gutaker et al. [51] identified a contemporary Chilean sample that carried a chlorotype from a wild relative used in potato breeding. López et al. [52] analyzed a wide subset of colored Chiloan potatoes from the UACh gene bank and detected accessions with DNA markers of the nuclear resistance *R*-genes/QTLs from wild species which were not involved in domestication. Nuclear SSR analysis revealed the genetic proximity of modern commercial varieties and the native potato cultivars collected on Chiloé Island between 2000 and 2007 [53]. These results may reflect genetic erosion resulting from spontaneous crosses between commercial varieties and Chilean landraces, as well as the replacement of landraces by modern varieties or breeding stocks. At the same time, it should be taken into account that the frequency of spontaneous crosses between introducents and Chilotanum could be limited due to the frequent male and/or female sterility of contemporary Chilean landraces [12,21,54].

Dodds [44] identified that the diversity of Chilean cultivated potatoes mainly remained in the Chilean germplasm collections. The potato gene bank in Chile was started after the expeditions of Montaldo in 1941 and 1942 and maintained at Centinela until 1958, when the Universidad Austral de Chile (UACh) took over responsibility for its maintenance. However, the main part of the Chilota Potato Genebank at the UACh represents material collected by Contreras in numerous expeditions in Chiloé and in adjacent territories conducted in 1969–2000 [55]; this germplasm was widely distributed. Contreras and Castro [55] presented a list of approximately 50 international and national potato-collecting missions that took place in Chile during the 20th century. Unfortunately, only a small proportion of them has been included in molecular studies. At present, several gene banks in Chile, Peru, Germany, Russia, Bolivia, and the USA maintain living germplasm collections of Chilean landraces [56].

The germplasm collections currently preserved in gene banks may not fully represent the genetic diversity of the Chilean landrace populations which existed before the late blight epidemic in Chile. Comparative molecular studies of contemporary gene bank accessions and historical herbarium specimens enable direct assessments of genetic changes that have occurred over time. This approach with genome-wide markers has recently been applied to Chilean potato [51], demonstrating that contemporary Chilean cultivars are very similar to modern European varieties and very different from the three historical Chilean samples collected by Darwin and Isern in the mid-19th century. However, the main limitation of studying genetic erosion in Chilean landrace populations is a very restricted number of historical herbarium samples [51].

The available herbarium specimens representing samples collected in Chile in the first decades of the 20th century are the most reliable material which records the diversity of Chilean cultivated potatoes before the appearance of *P. infestans*. Chilean potato specimens from the WIR Herbarium collected by the Russian botanist Juzepczuk (a colleague of Vavilov and Bukasov) can be attributed to such historical collections. Juzepczuk arrived in Chile at the beginning of February 1928 and collected potatoes mainly on Chiloé Island as well as in the surrounding areas until early July 1928 [57]. Based on this material, Bukasov described the intraspecific taxa of *S. tuberosum* s. str. [8]. Since that time, specimens of these taxa have been preserved at the Herbarium WIR in Saint Petersburg. In our study, we also included herbarium specimens of Chilean species that were considered by different authors as wild ancestors of Chilean cultivated potatoes: *S. maglia*, *S. leptostigma*, and *S. molinae*, as well as *Solanum ochoanum* Lechn. [58] and *Solanum zykinii* Lechn. [58] for completeness. Hawkes [3] classified the latter four species as synonyms of *S. tuberosum* subsp. *tuberosum*. Of the 61 herbarium sheets involved in the present study, half represent nomenclature types of potato species and their intraspecific taxa, i.e., 21 lectotypes, seven syntypes, one neotype, and one isolectotype [5,13,59,60].

To understand the genetic diversity of the Chilean landrace population collected by Juzepczuk in 1928, we analyzed authentic herbarium specimens of *S. tuberosum* from the WIR Herbarium with a widely used set of cpDNA markers for the detection of plastid DNA types [29] and with a set of 15 plastid SSR markers [36]. The results obtained were compared with data from the PCR analysis of living Chilean landrace accessions from the VIR field potato collection that were gathered in southern–central Chile after the appearance of *P. infestans* there.

## 2. Results

### 2.1. Geographic Origin of Authentic Herbarium Specimens

Based on the analysis of the herbarium label information, we can conclude that most (45 out of 61; 73.8%) herbarium specimens were collected on Chiloé Island (Appendix A, Figure 1). Out of 45 Chiloan specimens, 44 were collected at the following locations: 25 near Yutuy, five around Ancud, four at Quetalmahue, three at Cucao, three near Mechaico, three at Castro (in a farmer’s yard), and two samples on Chiloé Island with no specified settlement names. The remaining Chiloan specimen (k-7600) was collected by VIR scientist Zykin later, in 1967, on the western coast of Chiloé Island (about 43 °S), approximately 20–25 km south of Indian Cucao village [61].

Six landrace specimens were collected on the other islands. Four specimens were collected by Juzepczuk in 1928 on the island of San Sebastiano near Chiloé. Notably, the island of San Sebastiano is absent from modern topographic maps of South America; however, Bukasov [8] not only indicated this name for the island but also specified that it was “near Chiloé.” Modern maps show only Dona Sebastiana Island near Chiloé, slightly north of it. Perhaps there was a spelling mistake in the texts of the herbarium labels. Two specimens from Gran Guaiteca were passed to the VIR by Ochoa and described by Lekhnovich [58] as *S. ochoanum* Lechn.

Another six specimens were collected by Juzepczuk in 1928 on the mainland of southern Chile, namely, one at Temuco, two near Santiago, and three near Puerto Montt (Appendix A, Figure 1). In the herbarium label from a sheet of *S. leptostigma* (072), “Temuco” is indicated as the collection site. However, the protologue of this species does not mention this region and indicates only Chiloé, Cucao [11]. It is only known from the herbarium labels of four specimens (two of *S. molinae* and two of *S. maglia*) that the original plant material was collected in Chile (Appendix A).

### 2.2. Chlorotypes of Herbarium Specimens

#### 2.2.1. Plastid DNA Types of Herbarium Specimens

To identify the plastid DNA type in 61 herbarium specimens, PCR analysis was conducted with five plastid DNA-specific markers: H1 (T), S, SAC, A22, and H2 (Appendix A).

The H1 primers (T marker) from this set amplified two banding patterns in herbarium samples: a 202 bp fragment associated with the well-known 241 bp deletion in the *ndhC/trnV* intergenic spacer region (T-type cpDNA); and a 443 bp fragment without deletion in this region (Appendix A). The T-type cpDNA was present in 57 (95.9%) of 61 specimens of our subset; 4 samples lacked it (Table 1, Appendix A). The 57 herbarium specimens exhibiting the T-type cpDNA included 47 of 49 Chilean landrace samples (*S. tuberosum* s. str.), and all 10 specimens of *S. leptostigma*, *S. molinae*, *S. ochoanum*, and *S. zykinii* (Table 1 and Appendix A). The remaining four specimens lacking the 241 bp deletion included two landrace samples, namely, “Papa bastonesa” from Ancud (N 1940) and “Papa Bolera” from Yutuy (N 2002), and two specimens of wild species *S. maglia* (N 1831 and N 2871) (Appendix A).

Application of the SAC and the A22 markers developed in this study detected A-type cpDNA in three of the four herbarium specimens lacking the 241 bp plastid deletion (Table 1; Appendix A). Restriction reactions with the A22 and A primers for the fourth herbarium sample (oldest specimens in our subset of *S. maglia* of 1831) were unsuccessful. No samples with W-, C-, or S-type cpDNA were found in the set of herbarium specimens

The H2 primers amplified a single fragment in all 61 herbarium specimens. Two H2 marker bands were detected after the digestion of PCR products with restriction endonuclease *Hae*III in all 57 samples exhibiting the T-type cpDNA, whereas, in four specimens lacking plastid deletion, only an undigested 334 bp fragment was observed (Appendix A).

#### 2.2.2. Plastid Microsatellite Haplotypes of Herbarium Specimens

Fifteen plastid microsatellite primer pairs were amplified in all 61 herbarium specimens producing two cpSSR haplotypes, #II and #III; their allelic composition is presented in Appendix A. The dominant cpSSR haplotype #III was found in 57 (95.9%) holders of the 241 bp plastid deletion, including 47 out of 49 (95.9%) specimens of *S. tuberosum* s. str. and all 10 specimens of *S. leptostigma*, *S. molinae*, *S. ochoanum*, and *S. zykinii* (Table 1 and Appendix A).

Plastid SSR haplotype #II was found in the remaining four (4.1%) herbarium specimens possessing the A-type cpDNA—in two Chiloan landraces and in two samples of *S. maglia*, including the 1831 herbarium specimens from the Fisher collection [LE] (Table 1).

Combining all results of plastid DNA polymorphism analyses (cpDNA types and cpSSR haplotypes), two chlorotypes could be determined in the set of old herbarium specimens; their designations are given in Appendix A. In total, 57 (95.9%) of 61 herbarium specimens exhibited chlorotype cpT_III, and 4 (4.1%) samples possessed chlorotype cpA_II (Table 1 and Appendix A).

### 2.3. Chlorotypes of the Living Chilean Landrace Accessions from the VIR Field Potato Collection

The same five plasmid DNA-specific markers, H1 (T), S, SAC, A22, and H2, were applied to the living Chilean landraces (Appendix A). In a subset of 46 living accessions, 39 (84.8%) samples exhibited the 241 bp deletion (T-type cpDNA), and seven (15.2%) accessions lacked it (Table 2a and Appendix A).

The SAC and A22 markers classified these seven accessions into two groups with A-type cpDNA (two bands were generated both with the SAC and the A22 markers) and the W-type cpDNA (the SAC marker generated two bands, and the single band was observed with the A22 marker; Appendix A). Based on these results, the A-type cpDNA was identified in two living landrace accessions: k-7539 and k-7576. Type-W cpDNA was identified in five accessions: k-3414, k-5273, k-7535, k-7568, and k-7586 (Appendix A and Table 2a). No accessions with the S-type of cpDNA were found using the NTCP6 primer pair (S-marker) (Appendix A).

Similarly to the subset of herbarium specimens, restriction digestion with HaeIII yielded two fragments with the H2 marker in 39 living accessions exhibiting T-type cpDNA; fragmentation was not observed in seven accessions lacking the plastid deletion (Appendix A).

Thus, in the subset of 46 living landrace accessions from the VIR field potato collection, three of five plastid DNA types known for potatoes were found: T (84.8%), A (4.3%), and W (10.9%) (Table 2a).

Plastid microsatellite primers produced four cpSSR haplotypes, #II, #III, #V, and #Chl 3414; their allelic compositions are given in Appendix A. All 39 accessions with the 241 bp deletion exhibited the dominant cpSSR haplotype #III. Within the remaining seven accessions lacking this deletion, we detected three cpSSR haplotypes: #II (two samples), #V (four samples), and #Chl 3414 (one sample) (Appendix A).

Thus, four chlorotypes (Appendix A) were detected in the subset of 46 living landrace accessions from the VIR field potato collection: cpT_III (84.8%), cpA_II (4.3%), cpW_V (8.7%), and cpW_Chl 3414 (2.2%) (Appendix A).

### 2.4. Mitotypes and Cytoplasm Types of the Living Chilean Landrace Accessions from the VIR Field Potato Collection

Two mtDNA types, α and β, were detected in a subset of 46 living accessions based on the results of amplification with mtDNA primer pairs ALM_4/ALM_5. Five accessions possessing the W-type cpDNA had α-type mtDNA, whereas β-type mtDNA was detected in all thirty-nine accessions exhibiting the T-type cpDNA and in two landrace accessions exhibiting the A-type cpDNA (Table 2a and Appendix A).

The D (Region 1) primers specific to the *rps19* locus of mtDNA generated a single band 527 bp (the ‘Band 1′) in five landrace accessions (k-3414, k-5273, k-7535, k-7568, and k-7586) (Appendix A, Table 2a and Appendix A). It is known that the ‘Band 1′ marker is associated with cytoplasm type D derived from *S. demissum* [62,63]. All five accessions possessing the D-cytoplasm type had the W-type cpDNA and the α-type of mtDNA (Table 2a and Appendix A). All five accessions possessing the D-cytoplasm type had plastid cpSSR haplotype #V, with one exception—accession k-3414 possessed another combination of microsatellite alleles, which was identified previously as cpSSR haplotype #Chl 3414 [36] (Appendix A).

All three accessions of wild species *S. maglia* had the A-type cpDNA. Plastid SSR haplotype #IIs in these accessions were detected by us previously [36]. Hence, all three accessions of *S. maglia* had the cpA_II chlorotype, β-type mtDNA and A (A/β) cytoplasm type (Table 2a).

In total, two mitotypes, α and β, and three cytoplasm types, T (84.8%), A (4.3%), and D (10.9%), were found within the subset of 46 living landrace accessions from the VIR field potato collection. Table 2 summarizes our results, as well as data from the literature, which are discussed below.

### 2.5. Molecular Screening with Markers of the R1, R3a, and Ry_sto_ Genes of Living Chilean Landrace Accessions from the VIR Field Gene Bank

The whole subset of 46 living accessions of Chilean landrace cultivars was screened with PCR markers specific for the *R1* and *R3a* genes conferring race-specific resistance to late blight. The target fragments of these markers were detected in seven (15.2%) of 46 accessions; five of them had D-type cytoplasm (Appendix A). Thus, the R1 marker was identified in six samples (k-3414, k-7535, k-7543, k-7568, k-7586, and k-7599), and marker RT-R3a for the *R3a* gene was detected in one accession (k-5273). Two accessions (k-7543 and k-7599) exhibited the R1 marker, even though they had T-type cytoplasm (Appendix A).

None of the 46 living accessions of the Chilean landrace cultivars had the YES3-3B marker of the *Ry_sto_* gene.

In total, within the subset of 46 living Chilean accessions from the VIR field gene bank, which were collected after the appearance of *P. infestans* in Chile, the D-type cytoplasm derived from *S. demissum* was detected in five (10.9%) accessions and the diagnostic fragments of the markers of the *R1* or *R3a* genes—in 7 (15.2%) samples (Appendix A).

## 3. Discussion

Genetic data obtained from historic herbarium specimens greatly enhance the research on the domestication, introduction, and genetic diversity of crop plants [64]. Two papers are known for the molecular analysis of old herbarium specimens of *Solanum tuberosum*, both devoted to the history of European potato varieties and the change in their genetic diversity due to the late blight epidemics in Europe in 1848 [51,65]. Currently, not much information is available regarding the genetic diversity of Chilean landraces collected before the appearance of *P. infestans* in Chile, except for a limited set from three historic herbarium samples collected by Darwin and Isern in the 19th century in Chiloé and Chonos Archipelagos [51].

In our study, the largest sample size of Chilean landraces collected prior to the appearance of late blight in Chile was involved in genetic analysis. Our results document the cytoplasmic genetic diversity in landrace populations collected in southern–central Chile in 1928 based on the molecular study of 49 authentic herbarium specimens of *S. tuberosum* (*sensu stricto*) preserved in the WIR Herbarium and provide data on changes in their diversity after the appearance of *P. infestans* in Chile. We identified two cpDNA types (T and A), two cpSSR haplotypes (#III and #II), and two chlorotypes in the Chilean landrace population collected by Juzepczuk in 1928 with a clear predominance (96%) of chlorotype cpT_III, characterized by the presence of the 241 bp plastid deletion. The remaining 4% of herbarium specimens lacking this plastid deletion possessed chlorotype cpA_II. In the set of the living landrace accessions from the VIR field potato gene bank, the frequency of the dominant chlorotype cpT_III was 84.8%—significantly lower (*p*-value ≤ 0.05)—and, accordingly, the frequency of samples without this typical Chilean plastid deletion was significantly higher—15.2%, compared with the set of herbarium specimens.

Table 2b summarizes plastid DNA data from previous studies of contemporary Chilean landrace accessions maintained in four gene banks, although the sets of cytoplasmic DNA markers did not completely match in different studies. It turned out that the genetic diversity levels (frequencies of the Chilean cultivars possessing and lacking the 241 bp plastid deletion) did not differ significantly between the sets of living accessions from the VIR, UACh, NRSP-6, and CIP gene banks. However, all these estimates differed significantly (*p*-value ≤ 0.05) from the set of herbarium specimens representing the landrace population collected by Juzepczuk in 1928.

The main difference between the set of herbarium specimens and the set of Chilean landraces from the VIR field potato gene bank was in the appearance of living cultivars with the W-type cpDNA and cpSSR haplotype #V (chlorotype cpW_V).

To understand trends in genetic diversity changes in Chilean cultivars, we applied a set of commonly used markers specific to different loci of mitochondrial DNA [30,34,62] and demonstrated that all living Chilean accessions with the W-type cpDNA possessed the α-type mtDNA and the D-type cytoplasm which is characteristic for wild Mexican polyploid species, including *S. demissum* [62,63,66]. In addition, we showed that all living accessions with the D-type cytoplasm exhibited *R1*- or *R3a*-gene-specific markers. In the first part of the 20th century, potato breeding for late blight resistance mainly concentrated on the use of the wild Mexican species *S. demissum* as a source of major dominant race-specific genes (*R1*–*R11*) [67,68,69,70,71]. Commercial varieties exhibiting the *R*-genes from *S. demissum* had entered the market by the middle of the 20th century and, a few decades later, began to predominate within the breeding gene pool [67,68]. As expected, the frequency of varieties with D-type cytoplasm also increased in the second half of the 20th century [29,72,73]. Genotypes with D-type cytoplasm from *S. demissum* showed a higher average level of late blight resistance compared with those exhibiting T-type cytoplasm [73,74]. Therefore, it is highly probable that commercial varieties and breeding stocks derived from *S. demisum* were introduced to Chile after the appearance of *P. infestans*.

We also detected the allele-specific marker of the *R1* resistance gene in two living Chilean accessions exhibiting the T-type cytoplasm. It is possible that such genotypes could appear as a result of gene flow via pollen from the *S. demissum*-derived donor of the *R1* gene. While hybrids or varieties with *S. demissum*-derived D-type cytoplasm usually participate in crosses with *S. tuberosum* as maternal parents due to unilateral incompatibility [21,74,75,76] or functional pollen sterility [29]. However, more data are coming about finding efficient pollinators possessing the D-type cytoplasm [62,72,74,77] that could possibly be explained by the presence of functional alleles of the nuclear restorer genes [78,79]. At the same time, the detection of the *R1* gene marker is additional but not direct evidence of introgression from *S. demissum* because the markers of this gene have no clear phylogenetic signals; their homologous and pseudogenes were found in other potato species [80,81,82,83].

Núñez [50] applied PCR markers developed by Lössl et al. [30] for the set of 114 contemporary Chilean landrace accessions from the UACh gene bank and detected 14.9% of samples lacking the 241 bp plastid deletion and possessing the α type mtDNA that did not differ significantly from our data (Table 2b). The combination ‘absence of the plastid 241 bp deletion/α type mtDNA’ is not specific only to *S. demissum*; however, all holders of the D-type cytoplasm derived from *S. demissum* have such cytoplasmic combination [29,62]. It can be assumed that among the 14.9% of the samples found by Núñez, there were also genotypes with D-type cytoplasm. Unfortunately, at the time of Núñez’s 2007 study [50], the primers for the detection of D-type cytoplasm were not yet developed.

We consider genotypes with *S. demissum*-derived D-type cytoplasm as not native to Chile and assumed that they could have originated from spontaneous crosses between commercial varieties having Mexican species *S. demissum* in their pedigree and Chilean landraces or as the result of the replacement of native Chilean cultivars by such introducents.

*Solanum stoloniferum* Schltdl. is another wild Mexican species that has been widely used in breeding as a source of extreme resistance to PVY [67,69,70,71]. According to López et al. [52], 6.2% of contemporary samples collected in the Chiloé Province and preserved in the Chilota Potato Genebank (UACh) have the YES-3B marker of the *Ry_sto_* nuclear gene from *S. stoloniferum* [84]. In our subset of 46 living accessions, we did not find any samples with the YES-3B marker, probably because they were collected in Chile prior to the 1970s when commercial varieties with introgression from *S. stoloniferum* were not yet widely distributed.

The frequency of the Chilean landraces with the cpA_II chlorotype (A-type cpDNA, cpSSR haplotype II) was nearly the same (~4%) both in the set of herbarium specimens and in the set of living accessions from the VIR field potato collection. The occurrence of Chilean landraces with the -type cpDNA in the set from the NRSP-6 potato gene bank [26] did not differ significantly from the one we found (Table 2b). Chlorotype cpA_II is typical for *S. maglia* (present study), which was characterized by very rare fruiting. At the same time, most of the tetraploid Andean landraces also have the A-type cpDNA [26,34] as well as cpSSR haplotype II [36], while they are usually associated with male fertility and high fruiting ability.

It is interesting to note that Chilian landraces in which chlorotype cpA_II was found in our study were also characterized by a high fruiting ability known to us from the descriptions of the original samples. In the description of the authentic herbarium specimens of Bukasov [8], not only diagnostic morphological characters visible on the herbarium sheets are indicated, but also other additional information about flower and fruiting abilities, berry and tuber morphology. Thus, authentic herbarium specimen N 1940 (chlorotype cpA_II) belongs to *S. tuberosum* var. *multibaccatum,* which was distinguished by Bukasov [8] among the other Chilean samples based on the “very numerous berries and narrow leaflets.” According to the cultivar-group classification made by Kostina in 1978 [85], two living accessions, k-7539 and k-7576a (both with chlorotype cpA_II), belong to the cultivar-group Manteguila, which was characterized by a high fruiting ability [85].

It is possible that Chilean cultivars exhibiting the A-type cpDNA represent introductions of Andigenum that were misidentified in Chile as Chilotanum; they also can represent introductions to Chile of the 18th-century European varieties because most of them had Andean-type cpDNA [51,65]. Introductions of the contemporary commercial varieties with the cpA_II chlorotype in Chile are less probable because the A-type cpDNA is very seldom in the modern breeding gene pool [29,72,73,86]. In all cases, the varieties exhibiting Andean-type cpDNA must be able to tuberize under the long-day conditions of southern–central Chile. Our suggestions are consistent with the recent finding of the long-day-adaptive allele of the *StCDF1* gene in a set of the short-day-adapted Andigenum landraces and among old European varieties [38,51,71].

Subsequent crosses Andigenum x Chilotanum might be less probable due to the low pollen fertility/male sterility of the Chilean landraces reported by a number of authors [12,21,54,87]. However, the sources of pollen fertility restorer gene(s) were selected among breeding clones of *S. tuberosum* exhibiting the typical Chilean cytoplasm [74,88,89]. Several genomic fragments belonging to the *RFL-PPR* (restoration of fertility-like PPR) gene subfamily have been identified in genotypes exhibiting the T-type cytoplasm that might be considered *Rf* gene candidates in potatoes [79].

Notably, hybrids with T-type cytoplasm derived from reciprocal crosses Chilotanum × Andigenum (or from interspecific combinations where Chilotanum was used as the female) are characterized by different anther abnormalities and pollen sterility due to “nuclear–cytoplasmic male sterility” [21,90,91,92,93,94,95,96]. Therefore, the increase in the frequency of both highly fertile Andigenum landraces and male fertile genotypes having wild species introgression (breeding stocks and commercial varieties) leads to an increase in sterility in Chilean populations as a result of spontaneous crosses with such introducents as male parents. In this regard, we note that in the 1908 descriptions of landrace populations in southern Chile, given by Husbands [48], there is an indication of their fertility and the occurrence of numerous seedlings in indigenous fields. However, contemporary populations of Chilean *S. tuberosum* growing in the Chonos Archipelago lack flowers, fruits and seedlings [96].

Based on our results, one more aspect of the study of the diversity of Chilean potatoes should be discussed. Beginning with Darwin (cited from Ristano and Pfister [97]), a number of scientists who visited southern Chile have reported the existence of wild-growing potatoes [11,49,61,98,99]. The last reported finding of a wild-growing potato with long stolons came from the international expedition that was conducted in 1990 in the Guaitecas and Chonos Archipelagos [96]. Members of this expedition, including Contreras and Spooner, assumed that these beach populations of *S. tuberosum* may represent distinct biotypes different from the local landrace populations of southern Chile preserved in gene banks [96].

In our study, we found the same chlorotype cpT_III, which is typical for Chilean landraces, in eight authentic herbarium specimens of *S. leptostigma*, *S. molinae*, and *S. zykinii* which were collected on the uninhabited western coast of Chiloé Island and in two specimens of *S. ochoanum* from Grand *Guaiteca. This finding* does not contradict Hawkes’ assumption that these species are naturalized escapes from cultivation [3,12] or the conclusion of Contreras et al. about the existence of distinct biotypes of *S. tuberosum* [96].

## 4. Materials and Methods

### 4.1. Sampling of Chilean Herbarium Specimens

The present study comprised 61 Chilean potato herbarium specimens, of which 60 belonged to the WIR Herbarium maintained at the Vavilov Institute of Plant Genetic Resources (VIR), with 1 specimen belonging to the LE Herbarium of the Komarov Botanical Institute. Both herbaria are located in Saint Petersburg, Russia. Of the 61 herbarium specimens, 55 represent samples collected by Juzepczuk 94 years ago during his five-month expedition in Chile in 1928. These samples were transferred to VIR as tubers and planted at the VIR’s experimental station, “Krasnyj Pakhar,” near Leningrad (now Saint Petersburg). They were mainly herbarized in 1929, as evidenced by the text on the herbarium labels of 44 specimens (Appendix A). The remaining specimens represent samples collected by Zykin in Chiloé and by Ochoa in Gran Guaiteca. The oldest herbarium specimen in this subset is sample N 1831 of *S. maglia*, from the collection of Fisher of 1831, which is stored in the LE Herbarium.

Our subset included 49 herbarium specimens of *S. tuberosum*, 4 samples of *S. molinae*, 4 *S. leptostigma*, 2 *S. zykinii*, 2 *S. ochoanum*, and 2 specimens of *S. maglia* (Appendix A).

The 49 herbarium specimens of Chilean *S. tuberosum* belonged to the 8 botanical varieties and 16 forms that Bukasov [8] listed in their original description of intraspecific taxa of *S. tuberosum* s. str. (Appendix A). The reality of the existence of these intraspecific taxa can be accepted or not; however, it should be recognized that their names were validly published in accordance with Art. 6, Art. 39 of the *International Code of Botanical Nomenclature of Fungi, Algae, and Plants* [100]. All these intraspecific taxa of *S. tuberosum* were supplied with complete diagnoses in Russian and with reference to the authentic specimens (expedition number and collecting site), which were used to describe them [8]. In accordance with Article 9.6, all specimens that were specified in the original description of Bukasov [8] can be attributed to syntypes.

There were 44 out of 49 specimens of *S. tuberosum* herbarized in 1929 (Appendix A). The remaining 5 specimens of *S. tuberosum* are represented by plants from the next tuber reproductions, which were collected for the herbarium in 1930 (1 sample), 1931 (3 samples), and 1936 (1 specimen; Appendix A). Thus, 49 herbarium specimens of Chilean *S. tuberosum* ranged in absolute age from 86 to 93 years old.

The situation with two herbarium specimens of *S. leptostigma* and 4 specimens of *S. molinae* is more complicated. According to the protologues, these samples were grown from tubers that were collected near the Indian village of Cucao, Chiloé Island, and handed to Juzepczuk by Junge (samples of *S. leptostigma*) and by Robert Christie (samples of *S. molinae*) [11]. However, this information is missing from the herbarium labels. Plants of the 2 specimens of *S. molinae* were from the third tuber reproduction of 1931; 2 other sheets of this species represent plants from the later tuber reproductions of 1954 and 1963 (Appendix A).

Four herbarium specimens of *S. zykinii* and *S. ochoanum* represent plant material collected in Chile in the second half of the 20th century. One herbarium sheet of *S. zykinii* represents a specimen collected by a VIR researcher, Zykin, in Chiloé in 1967—this is a plant of the VIR accession k-7600 that was herbarized in 1978. Another specimen of *S. zykinii* collected by Ochoa was herbarized in 1968 from the plant of the accession k-11288, which was grown in the VIR field collection. Two herbarium sheets of *S. ochoanum* were prepared in 1978 from samples k-11289 and k-11290 grown in the VIR field collection from tubers collected by Ochoa on the Guaitecas Archipelago and passed to VIR (Appendix A).

Our study also involved two herbarium sheets of *S. maglia*: one specimen from the old collection of Fisher of 1831, which is stored in the LE Herbarium; the second sample represents VIR accession k-2871, reproduced in 1954 (Appendix A).

To avoid complicating taxonomic history and synonymy, we not only refer to the classification system of Russian taxonomists [7,8,11,58] but also to the taxonomic treatment of Hawkes [3] and the latest classification system developed by Spooner et al. [4] and Ovchinnikova et al. [5].

### 4.2. Sampling of Living Chilean Accessions from the VIR Field Gene Bank

In addition to the subset of herbarium specimens, 46 living accessions of the Chilean landraces of *S. tuberosum* were included in our study; these samples were obtained from the potato field gene bank of the Vavilov Institute of Plant Genetic Resources (VIR) (Appendix A). These accessions represent samples collected in Chile after the appearance there of *P. infestans*. Of the 46 accessions, 32 came from the VIR expeditions in southern–central Chile led by Zhukovsky in 1958 and by Zykin in 1967. An additional 14 accessions arrived at the VIR from foreign gene banks: Commonwealth Potato Collection (CPC), UK; the German gene bank, Gross Lüsewitz; and from the NRSP-6 Potato Genebank, USA (Appendix A). However, the main part of this subset represents material originally derived from the Chilean potato gene bank. Three samples of wild *S. maglia* species (PI 245087, PI 558315, and PI 558316) from the NRSP-6 were also involved in our study.

### 4.3. Geographical Distribution of Chilean Herbarium Specimens

We used Map Info (version 9.5) software to construct a map of the collection sites of the 61 Chilean herbarium specimens. Coordinates of the collection sites could not be established for 4 of the 61 herbarium specimens due to the incomplete geographical information in herbarium labels.

### 4.4. DNA Isolation

Limited quantities (0.025–0.040 g) of leaf tissue were selected from the 61 herbarium specimens by the curator of the WIR potato herbaria. DNA was isolated following the modified CTAB method [36]. To reduce the polyphenol compounds partially oxidized in the herbarized plant tissues, the extraction buffer contained a higher amount of 2-mercaptoethanol (2%). In some cases, DNA samples were additionally purified using silica membrane mini columns (Invitrogen kit) or with silica-coated magnetic beads (MiniPrep kit by Sileks Co., Moscow, Russia); in both cases, we followed the manufacturers’ protocols.

The same method [36] was used for DNA isolation from the fresh leaves of living plants grown in the VIR field gene bank. In this case, at the initial stage, leaf tissues were ground in liquid nitrogen.

### 4.5. Determination of Organelle DNA Type in Chilean Specimens

A commonly used set of plastid DNA and mitochondrial DNA markers (STS, CAPS, and cpSSR) recommended by Hosaka and Sanetomo [29] was applied to identify cpDNA, mtDNA, and cytoplasm types of Chilean specimens (Appendix A). For most markers of this set (T(H1), NTCP6, SAC, S, and D), PCR analysis was reliable for both the old herbarium specimens and for living accessions. However, PCR with the ALM4/ALM5 and the A primers in the case of the old herbarium material was unsuccessful. We assumed that the amplification of relatively long PCR products (expected sizes of 1600 bp for ALM4/ALM5 and 1178 bp for A primers) was problematic on degraded DNA, which was isolated from tissues of old herbarium plants.

For the detection of A-type cpDNA, we developed new specific primer pair, A22, which generated a relatively short PCR product (Appendix A, Appendix A). For this, we detected the annealing sites for primers A recommended by Hosaka and Sanetomo [29] in the potato cpDNA sequence DQ231562.1 (cv. Desiree, T-type). In the corresponding PCR product, we identified the site GGATCG in which the base change to GGATCC formed an extra BamHI recognition site specific to the A-type cpDNA (Appendix A). New A22 primers were developed using Primer3Plus software (https://primer3plus.com, accesed on 26 November 2022). Primer pair A22 generated an 805 bp PCR product, which is shorter compared with amplicons of the A primers (1178 bp) and the other product created by Mihovilovich et al.—1020 bp [74]—and could successfully amplify the DNA from old herbarium material.

In addition, an H2 marker was included in our study, which, according to Hosaka [32], can detect rare haplotypes among *S. tarijense* accessions exhibiting T-type cpDNA.

Thus, for the herbarium material, we were able to determine only the plastid DNA types, whereas, for the living accessions, we detected both cpDNA and mtDNA types; using their combination enables the determination of potato cytoplasm types [29].

PCRs were performed in a 20 µL volume containing 40 ng of genomic DNA, 1 × reaction buffer with 2.5 mM MgCl_2_, 0.6 mM of each dNTP, 500 pM each of forward and reverse primer, and 1 unit of Taq polymerase. The PCR conditions, in general, corresponded to data presented in the literature (Appendix A). PCR products of SAC and A22 markers were digested with the restriction endonuclease BamHI, and the PCR product of H2 was digested with HaeIII restrictase (Appendix A). Restriction enzymes for the CAPS markers were produced at SibEnzyme. The reactions were performed overnight, adhering to the manufacturer’s protocol.

The PCR and restriction products were separated in 2% agarose gels in a TBE buffer stained by ethidium bromide and visualized under UV light.

### 4.6. Plastid Microsatellite Analysis

We applied the previously published set of plastid SSR primer pairs (cpSSRs) to amplify mononucleotide repeats in the single copy regions of the plastid genome of herbarium specimens and living accessions. This set included 12 cpSSRs developed by us [36] and 3 cpSSRs developed by Bryan et al. [101] (Appendix A). PCRs were performed in a 12 µL volume containing 40 ng of genomic DNA, 1× reaction buffer with 2.5 mM MgCl2, 0.4 mM of each dNTP, 200 pM of M13-tailed forward SSR primer, 200 pM of reverse SSR primer, 25 pM of 700 or 800 IRDye-labeled M13 forward primer (cacgacgttgtaaaacgac), and 1 unit of Taq polymerase (Dialat, Moscow, Russia). For more stringent PCRs, the following programs with a touchdown profile were used: 3 min at 94 °C, 5 cycles [45 s at 94 °C, 1 min at T°m + 5 °C with a decrease of 1 °C per cycle, 1 min at 72 °C], 5 cycles [45 s at 94 °C, 1 min at T°m, 1 min at 72 °C], 30 cycles [45 s at 94 °C, 45 s at T°m, 1 min at 72 °C], with a final extension step of 5 min at 72 °C.

In the microsatellite assay, PCR products were separated in a 6.5% denaturing polyacrylamide gel using an LI-COR DNA Analyzer 4300S unit, following the procedure proposed by the manufacturer. Electrophoresis was performed in 41 cm long gels, enabling certain diagnoses with one-nucleotide differences in fragment lengths. Molecular weight markers of 50–350 bp IRD 700/800 (Li-COR #4200-44/#4000-44B) were taken as standards. As a control, DNA of the potato variety Desiree (T-type of cpDNA) was used; in this case, the sizes of the PCR products could be determined by analyses of complete sequences of the chloroplast genome (GenBank #DQ231562.1).

The names of the cpSSR haplotypes corresponded to our previous research [36], where the set of the same 15 primers was used to study the genetic diversity of 392 living accessions of cultivated species and their wild relatives.

In the present study, plastid SSR haplotypes were determined for the first time for all 61 herbarium specimens and for 22 living Chilean landrace accessions from the VIR field potato collection. The plastid SSR haplotypes of 24 living Chilean landrace accessions and of 3 accessions of *S. maglia* were determined in our previous research [36].

### 4.7. Detection of the Diagnostic Markers of the Resistance R-Genes from Mexican Polyploid Species

In the set of living accessions, we used the gene-specific markers R1 [102,103] and RT-R3a [104] to detect the *R1* and *R3a* genes, conferring race-specific resistance to late blight pathogen. Besides, we used marker YES3-3B [84,105] for detecting the gene *Ry_sto,_* which confers extreme resistance to potato virus Y (PVY) (Appendix A).

## 5. Conclusions

The first contradictory hypotheses of the origin of Chilean potatoes (*S. tuberosum Chilotanum* group) have been revised in recent decades. Genetic data indicate that the Chilean cultivated potato is not simply an Andigenum representative selected for its ability to form tubers under long day conditions, but it has evolved through several rounds of interspecific hybridization and introgression with different wild Bolivian and Argentinian species from the primary potato gene pool which do not belong to the *S. brevicaule* domestication complex [33,38,40]. The broadened genetic diversity of these hybrid populations has made it possible to select genotypes with photoperiod adaptation allele(s) that were able to spread to new geographic and ecological areas, such as lowland southern Chile with its long day conditions [38,51,71]. Theoretically, a low level of genetic variation (including cytoplasmic genetic diversity) can be expected in the “Pre-Chilotanum” populations who migrated to southern Chile in prehistoric times due to founder (bottleneck) effects.

Although the issue of the origin of Chilean potatoes is being actively discussed and investigated, the question of the changes in its genetic diversity during the 20th century (before and after the late blight epidemics in Chile) has seldom been studied due to the limited number of specimens collected before the disease outbreaks. This issue was the subject of our investigation. Almost all (96%) herbarium specimens from the Chilean landrace population of 1928 had the same cpT_III chlorotype. Finding the new chlorotypes within subsets of contemporary living Chilean accessions that were not detected within Chilean landraces collected in 1928 (before the late blight epidemics in Chile) might indicate genetic erosion through the introduction/introgression of genetic material from commercial varieties having in their pedigree wild species of the secondary potato gene pool. Moreover, detection of the D-type cytoplasm in living Chilean accessions typical only for Mexican polyploids [62,63,66] indicates introgression from wild Mexican species *S. demissum* belonging to the secondary gene pool. It is important to note that potato varieties that were developed in the 20th century represent high genetic diversity resulting from the introduction of many introgressions of wild species both from the primary and secondary potato gene pools [67,70,71]. Using a modern genomic approach, Hardigan et al. [38] revealed extensive allele sharing between landraces, wild species, and North American commercial varieties. Cross-pollination between commercial varieties and Chilean landraces remaining after epidemics was likely. Thus, extensive gene flows were demonstrated previously in Andean field plots [106,107,108]. Our results confirm that genetic erosion could lead not only to a decrease but also to an increase in crop genetic diversity [108,109].

Finally, one major problem hindering investigations of the origin and genetic diversity of Chilean potato is the limited availability of plant material that was collected in southern Chile before the late blight epidemic. Our results provide an example of the value of old herbarium samples for the analysis of genetic erosion in landrace populations. It would be interesting to extend these data using genomic approaches.

## Figures and Tables

**Figure 1 plants-12-00174-f001:**
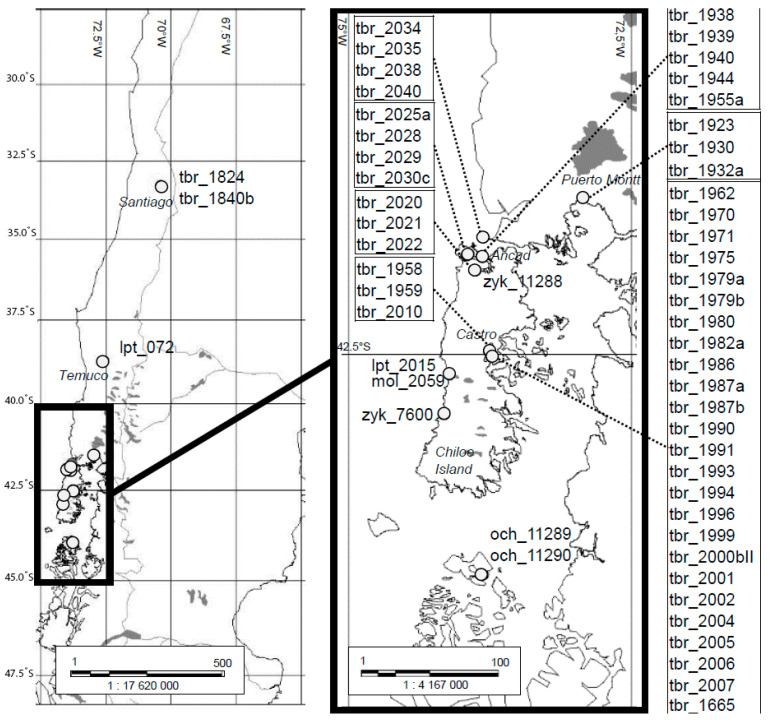
Map showing the collection sites of Chilean samples from the herbarium subset. Tbr, *S. tuberosum* subsp. *tuberosum* (*S. tuberosum* ‘Chilotanum group’; lpt, *S. leptostigma*; mol, *S. molinae*; och, *S. ochoanum*; *zyk*, *S. zykinii* (see also Appendix A).

**Table 1 plants-12-00174-t001:** Frequency of plastid DNA types and cpSSR haplotypes in Chilean potatoes based on the PCR analysis results of 61 herbarium specimens.

Name of Species	Number of Analyzed Herbarium Samples	Number (Frequency) of Specimens with Different		Name of Species According to			Number of Analyzed Herbarium Samples	Number (Frequency) of Herbarium Specimens with Different cpDNA Types and cpSSR Haplotypes
		cpDNA Type	Plastid SSR Haplotypes	Hawkes, 1990 [3]		Spooner et al., 2007; Ovchinnikova et al., 2011 [4,5]		
Chilean cultivated potato				Chilean cultivated potato				
*S. tuberosum* L. (*S. tuberosum* s.str.)	49 ^1^	T—47 (0.96) A—2 (0.04)	#III—47 (0.96) #II—2 (0.04)	$\begin{array}{l} \;\\ \;\\ \;\\ \;\\ \;\\ \;\\ \;\\ \;\\ \;\\ \;\\ \;\\ \;\\ \;\\ \;\\ \;\\ \; \end{array}\}$	*S. tuberosum* subsp. *tuberosum*	*S. tuberosum* Chilotanum Group	59	T (#III)—57 (0.97) A (#II)—2 (0.03)
Wild Chilean species according to Russian taxonomists:								
*S. molinae* Juz.	4 ^1^	T—4	#III—4					
*S. leptostigma* Juz.	2 ^1^	T—2	#III—2					
Total number of samples collected in Chile in 1928	55 ^1^	T—53 (0.96) A—2 (0.04)	#III—53 (0.96) #II—2 (0.04)					
*S. ochoanum* Lechn.	2 ^2^	T—2	#III—2					
*S. zykinii* Lechn.	2 ^2^	T—2	#III—2					
Wild species: *S. maglia* Schltdl.	2	A—2	#II—2	Wild species: *S. maglia* Schltdl.			2	A (#II)—1 ^3^
Total number of herbarium specimens	61						61	

^1^ herbarium specimens representing plant material collected in 1928; ^2^ herbarium specimens representing plant material collected in the second half of the 20th century (in 1967 and in 1978); ^3^ restriction reactions with the A22 and A primers for one herbarium sample of *S. maglia* (1831 specimen from the Fisher collection [LE]) were unsuccessful, although cpSSR primers were successfully applied to this old herbarium sample.

**Table 2 plants-12-00174-t002:** Frequency of cpDNA, mtDNA haplotypes, and cytoplasm types that were determined in the living accessions of *S. tuberosum Chilotanum Group* (=*S. tuberosum* subsp. *tuberosum*) maintained in the VIR field gene bank (present study) and different gene banks (literature data).

Gene Bank Collection—Source of Analyzed Accessions (Reference)	No. of Analyzed Accessions	Number (Frequency, %) of Accessions with Specific Organelle DNA Haplotypes				Nomenclature of Cytoplasm Types of Lössl et al., 2000 ^1^ [30]
		cpDNA with the 241 bp Deletion/without This Deletion	cpDNA-Type	mtDNA-Type	Modern Nomenclature of Potato Cytoplasm Types of Hosaka and Sanetomo, 2012 [29]	
(**a**) **Present Study**						
VIR, Russia VIR field gene bank Potato collection (Present study)	46	with deletion—83.0	T-type—84.8	β-type—89.1	T (T/β)—84.8	T β—84.8
		without deletion—17.0	A-type—4.3		A (A/β)—4.3	without deletion/β—4.3
			W-type—10.9	α-type—10.9	D (W/α)—10.9	without deletion/α—10.9
			other types of cpDNA—0	other types of mtDNA—0	other cytoplasm types—0	
(**b**) **Data from the Literature**						
UACH, Chile [50]	114 ^2^	with deletion—80.7	T-type –80.7	β-type—82.5	n.d.	T/β—78.1
				α-type—17.5		T/α—2.6
		without deletion—19.3	other types of cpDNA—n.d.			without deletion/β—4.4
				other types of mtDNA—0		without deletion/α—14.9
NRSP-6, USA [26]	24	with deletion—87.5	T-type–87.5	n.d.	n.d.	n.d.
		without deletion—12.5	A-type—8.3			
			W-type—4.2			
			other types of cpDNA—0			
NRSP-6, USA [29,34]	8	with deletion—87.5	T-type—87.5	β-type—100	T (T/β)—87.5	T/β—87.5
		without deletion—12.5	A-type—12.5		A (A/β)—12.5	without deletion/β—12.5
			other types of cpDNA—0	α-type—0	0	
CIP, Peru [4]	27	with deletion—81.5	T-type—81.5	n.d.	n.d.	n.d.
		without deletion—18.5	other types of cpDNA—n.d.			
CIP, Peru [35] Supplementary Table	134 ^3^	with deletion—82.1	T-type—82.1	n.d.	n.d.	n.d.
		without deletion—17.9	other types of cpDNA—n.d.			

“n.d.”—not determined; ^1^ to compare our results with data from the literature from the UACH collection, we also used the old nomenclature of cytoplasm types of Lössl et al., 2000 [30], which is not currently used; ^2^ Núñez 2007 [50] analyzed 129 accessions preserved in the UACH, 114 of which are Chilean landrace cultivars, and it is for them that we present the estimations here; ^3^ the whole subset of the *S. tuberosum* Chilotanum Group obtained by CIP from different sources was 190 accessions; however, for present estimation, we selected 134 accessions collected in the Chonos Archipelago.

## Data Availability

Not applicable.

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
