# Peer review of "Comparative Analysis of the Genetic Diversity of Chilean Cultivated Potato Based on a Molecular Study of Authentic Herbarium Specimens and Present-Day Gene Bank Accessions"

_plants, 2022, doi:10.3390/plants12010174_

Round 1
Reviewer 1 Report
Please see the attachment.

Author Response
We are grateful to Reviewer 1 or valuable critical comments. Please find our responce on your comments in the attachment.

Reviewer 2 Report
See attached file.

Author Response
We are grateful to Reviewer 2 or valuable critical comments. Please find our responce on your comments in the attachment.

Round 2
Reviewer 1 Report
The revised MS by Gavrilenko et al. reveals many excelent improvements, especially to the Abstract, Discussion and Conclusion. Nevertheless, several points raised in my previous review have not neen properly attended. They include the references, the protocol for DNA isolation and PCR analysis (lack of proper controls) and insufficiently founded conclusions based on the markers of R genes. There are vast possibilities for further editing the MS. However, the quality of the article is the responsibility of the authors, not the reviewers. Therefore I am ready to finally conclude that the manuscript has been sufficiently improved to warrant publication in Plants.